# Bacterial Ghosts of *Pseudomonas aeruginosa* as a Promising Candidate Vaccine and Its Application in Diabetic Rats

**DOI:** 10.3390/vaccines10060910

**Published:** 2022-06-07

**Authors:** Salah A. Sheweita, Amro A. Amara, Heba Gamal, Amany A. Ghazy, Ahmed Hussein, Mohammed Bahey-El-Din

**Affiliations:** 1Department of Clinical Biochemistry, Faculty of Medicine, King Khalid University, Abha 62529, Saudi Arabia; 2Department of Biotechnology, Institute of Graduate Studies & Research, Alexandria University, Alexandria 21568, Egypt; hebagamal983@yahoo.com (H.G.); ahmed.hussein@alexu.edu.eg (A.H.); 3Protein Research Department, Genetic Engineering and Biotechnology Research Institute, City of Scientific Research and Technological Applications, New Borg El-Arab, Alexandria 21934, Egypt; amaraamro@yahoo.com; 4Department of Microbiology & Medical Immunology, Faculty of Medicine, Kafrelsheikh University, Kafrelsheikh 33516, Egypt; ghazy.amany@yahoo.com; 5Department of Microbiology and Immunology, Faculty of Pharmacy, Alexandria University, Alexandria 21568, Egypt; m.bahey-el-din@alexu.edu.eg

**Keywords:** *Pseudomonas aeruginosa*, bacterial ghost, vaccine, diabetic ulcer, sponge-like reduced protocol (SLRP), oral immunization

## Abstract

Infections with *Pseudomonas* *aeruginosa* (PA) pose a major clinical threat worldwide especially to immunocompromised patients. As a novel vaccine network for many kinds of bacteria, bacterial ghosts (BGs) have recently been introduced. In the present research, using Sponge-Like Reduced Protocol*, P. aeruginosa* ghosts (PAGs) were prepared to maintain surface antigens and immunogenicity. This is the first study, to our knowledge, on the production of chemically induced well-structured bacterial ghosts for PA using concentrations of different chemicals. The research was carried out using diabetic rats who were orally immunized at two-week intervals with three doses of PAGs. Rats were subsequently challenged either by the oral route or by the model of ulcer infection with *PA*. In challenged rats, in addition to other immunological parameters, organ bioburden and wound healing were determined, respectively. Examination of the scanning and transmission electron microscope (EM) proved that PAGs with a proper three-dimensional structure were obtained. In contrast to control groups, oral PAGs promoted the generation of agglutinating antibodies, the development of IFN-γ, and the increase in phagocytic activity in vaccinated groups. Antibodies of the elicited PAGs were reactive to PA proteins and lipopolysaccharides. The defense against the PA challenge was observed in PAGs-immunized diabetic rats. The resulting PAGs in orally vaccinated diabetic rats were able to evoke unique humoral and cell-mediated immune responses and to defend them from the threat of skin wound infection. These results have positive implications for future studies on the PA vaccine.

## 1. Introduction

*P. aeruginosa* is a Gram-negative opportunistic pathogen that can cause community and hospital-acquired infections with harmful effects ranging from cosmetic to life-threatening infections [1]. It produces widespread and often overwhelming infections such as skin/soft tissue infections, ulcerative keratitis, diabetic wound and foot infection [2,3], infection of burn wounds, urinary tract infections, bacteraemia, and pneumonia [4,5]. PA is accountable for 11–13.8% of all hospital-acquired infections with a high prevalence of exotoxin production along with antimicrobial resistance. This situation is alarming and requires immediate interventions to develop new treatments and prevention policies against PA infections in hospitals [4,5]. Many attempts have been made to obtain an effective vaccine against *P. aeruginosa* [6]. However, no vaccine is available so far due to efficacy and/or safety concerns [6].

Bacterial ghosts (BGs) are empty non-deformed bacterial envelopes that are devoid of cytoplasmic content and possess all bacterial bio-adhesive surface properties in their original state. They can be produced by controlled expression of cloned gene E from bacteriophage, forming a lysis tunnel structure within the envelope of the living bacteria [7,8]. Using the E lysis gene method to prepare BGs causes efficient but incomplete lysis/killing of the host bacteria. This necessitates the use of chemicals such as β-propiolactone in a subsequent step to kill the remaining living bacteria [9]. Furthermore, using cloned or genetically modified elements (*E* lysis gene) for BGs preparation may raise concerns and rigorous quality control requirements in regulatory bodies for obtaining clinical use licensure.

We previously prepared BGs using Plackett-Burman experimental design approach to identify the best conditions resulting in well-structured immunogenic BGs [10]. The latter method, called the “sponge-like” or SL method, tested 12 experimental conditions of various chemical concentrations to create *Escherichia coli* BGs [10]. We further selected the best two experimental conditions that resulted in high-quality BGs and used these conditions as universal experiments for the preparation of BGs of different Gram-negative bacteria. The latter method using a reduced number of experiments was called “sponge-like reduced protocol” or SLRP [11]. In SLRP, critical concentrations of some chemical compounds are used to create BGs. These chemicals include sodium dodecyl sulfate (SDS) and sodium hydroxide (NaOH), which have been reported for their ability to interfere with the bacterial cell wall, and hydrogen peroxide (H_2_O_2_) which is well known for its ability, as an oxidant, to degrade DNA [10,11].

More recently, we have reported that *Klebsiella pneumoniae* BGs, prepared by SLRP, can be used as vaccine candidates with intrinsic adjuvant properties grounded on well-known immune-stimulating pathogen-associated molecular patterns (PAMPs) [12].

In the present study, we used, for the first time, the Sponge-Like Reduced Protocol to prepare *P. aeruginosa* ghosts (PAGs) keeping microbial surface antigens and evaluated their ability to induce specific immune response. The oral route of immunization was investigated in a diabetic rat infection model.

## 2. Material and Methods

### 2.1. Reagents, Bacterial Strain, and Cultivation

All reagents used in the study were supplied by Sigma–Aldrich Chemical (St Louis, MO, USA) unless otherwise stated. The *P. aeruginosa* clinical strain used in this study has been kindly provided by the Department of Microbiology, Faculty of Medicine, Tanta University, Egypt. The purity of the strain was confirmed by colony morphology on MacConkey agar, cetrimide agar, and blood agar. Identity was confirmed using the automated microbial identification system VITEK^®^ 2 (Biomerieux, Marcy-étoile, France).

### 2.2. Preparation of P. aeruginosa Ghosts (PAGs)

PAGs were prepared by the Sponge-Like Reduced Protocol as previously described [11]. The concentrations that exhibited first growth after the MIC for each of NaOH, SDS, H_2_O_2_, and CaCO_3_ were determined using the twofold dilution method technique in nutrient broth (NB) medium, as previously described [10,11,13]. Tubes were incubated overnight, and both MIC and the MGC for each chemical were determined. The MIC of NaOH, SDS and H_2_O_2_ were 0.5 mg/mL, 0.05 mg/mL, and 0.015%, respectively. The MGC was used as one-tenth the determined MIC concentration. For CaCO_3_, MIC and MGC were not determined and it was used at high and low concentrations of 1.05 and 0.35 µg/mL, respectively, as previously described [11]. Viability assessment of PAGs was done by culture on nutrient medium which revealed no growth and confirmed the complete killing of PA cells. Smears of cells of both PAGs and PA were stained with crystal violet and compared with stained live PA under light microscope (LM).

PAGs were prepared using three consecutive steps: basic step, H_2_O_2_ step, and ethanol step. Briefly, PA cells were first incubated with a mixture containing either the MIC or MGC of NaOH, SDS, and CaCO_3_ for 1 h followed by centrifugation and washing with sterile saline. The cells were then incubated with MIC or MGC of H_2_O_2_ for 30 min. Again, following centrifugation and washing with sterile saline, the pellet was treated with 60% ethanol for 30 min followed by centrifugation and resuspension in sterile saline. Two experimental schemes were tried for PAGs preparation (Table 1). The quality of PAG preparations was determined using light microscope and scanning electron microscope. DNA and protein concentrations were determined spectrophotometrically at 260 and 280 nm, respectively. Agarose gel electrophoresis was performed to show DNA content in both living bacteria and PAGs. DNA samples from each PAG preparation were checked for the existence of any viable cells [10].

### 2.3. Animals and Experimental Design

Twenty-four male Sprague-Dawley rats, weighing 70 ± 10 g, were purchased from the animal house of the Faculty of Medicine, Alexandria University, Egypt. Animals were handled in accordance with the principles of laboratory animal care, National Institutes of Health (NIH) guide for laboratory animal welfare. The rats were maintained at a temperature of 25 ± 2 °C, relative humidity of 40–60%, with a 12 h light/dark cycle and access to a pellet diet and water ad libitum. All animal procedures were approved by the animal care and use ethics committee of Alexandria University.

After two weeks of acclimatization, rats were divided into four groups, six rats each. Rats were rendered diabetic by intraperitoneal (IP) injection of alloxan (single dose of 120 mg/kg) [14]. Fasting blood glucose levels were measured in rats after two days where rats exhibited a blood glucose level over 200 mg/dL.

Two groups of diabetic rats were then immunized via the oral route with 100 µL containing 10^8^ PAGs in sterile saline on days 1, 14, and 28. The other two groups of diabetic rats served as a negative control and were given oral saline.

### 2.4. Oral Challenge with P. aeruginosa Clinical Strain

One PAGs-immunized diabetic group and one negative control diabetic group were challenged orally on day 35 with 10^8^ colony forming units (CFU) of the clinical PA strain. The rats were vaccinated orally using blunt-ended oral gavage needle. Five days later, rats were euthanized, and organ bioburden was determined. Briefly, isolated organs were weighed, homogenized, serially diluted in sterile saline, and plated onto cetrimide agar. After 24 h incubation at 37 °C, PA colonies were counted and bioburden per gram organ was calculated.

### 2.5. Preparation of Artificial Ulcers Followed by Topical PA Challenge

Rats of the other two remaining groups, both PAGs-immunized and negative control, were shaved on day 35, and residual hair removed by Veet hair removal cream (Reckitt and Colman Pharmaceuticals Co., Giza Governorate, Egypt), in an area of 5 cm^2^. After complete removal of skin hair, 10 μL keratolytic solution (October Pharma Co., Giza Governorate, Egypt), consisting of salicylic acid (0.2 g/mL), lactic acid (0.05 g/mL), and Palidocanol (0.02 g/mL), were spotted on the surface of 0.5 cm^2^ filter papers that were tightly attached to the rats’ skin by the aid of silk tapes and left for 20 min or until skin bleeding occurred. The artificial ulcers were subsequently infected with PA using a cotton swab as previously described [15]. The process was repeated three times at one-hour intervals to make sure that the ulcers were infected [15]. Rats were observed for 6 days for survival and ulcer healing.

A schematic representation is shown in Figure 1 to summarize the animal experimental design used for the oral and ulcer challenge.

### 2.6. Slide Agglutination Test for Detection of P. aeruginosa-Specific Antibodies following Oral Immunization with PAGs

Serum samples were collected from all rats on day 34 of the vaccination regimen. Agglutinating antibody titers, as an indicator for humoral immune response raised against PA, were measured in serum samples of all studied rats by slide agglutination test according to the method described by Sheweita and colleagues [16]. Briefly, serial dilutions of rat serum were mixed with PA homogenous suspension on a glass slide. The suspension was subsequently observed for clumping/agglutination for 1–3 min. The highest serum dilution that resulted in agglutination was taken as the antibody titer. Each sample was done in triplicates and the mean titer was calculated.

### 2.7. Evaluation of Phagocytic Activity

Phagocytic activity of polymorphonuclear cells (PMNs), isolated from vaccinated and control rats on day 34, was investigated against living PA using the acridine orange method as previously described [16,17]. Briefly, PA suspension (2 × 10^8^ CFU/mL) was mixed with an equal volume of serum at 37 °C for 30 min with continuous gentle shaking. The resulting opsonized bacteria were pelleted and resuspended in sterile saline. PMN were subsequently added to get a multiplicity of infection (MOI) of 1000 bacteria per each PMN cell. The latter reaction was incubated at 37 °C for 30 min with gentle shaking after which the reaction was centrifuged and the pellet was stained with 200 µL acridine orange (15 mg/L) for 1 min followed by saline washing. PMNs were examined under a fluorescent microscope using 40× objective lens. Phagocytic activity was expressed as the percentage of PMNs containing two or more ingested organisms from the total number of examined PMNs using the following equation:$$\mathrm{Percentage}\;\mathrm{phagocytic}\;\mathrm{activity}=\frac{\mathrm{number}\;\mathrm{of}\;\mathrm{ingesting}\;\mathrm{PMNs}}{\mathrm{total}\;\mathrm{number}\;\mathrm{of}\;\mathrm{PMNs}}\times 100$$

### 2.8. Nitro-Blue Tetrazolium Test (NBT)

Heparinized blood samples collected on day 34 from vaccinated and unvaccinated rats were incubated with a buffered solution of NBT. Smears were then prepared, stained, and examined microscopically to determine the percentage of neutrophils showing intracytoplasmic deposits of insoluble formazan (a reduced form of NBT) [12,18]. At least five different fields were examined per slide and percentage NBT reduction was calculated. Neutrophils with single large black cytoplasmic deposits of formazan were considered as positive [18]. The formation of insoluble formazan reflects the killing activity of neutrophils.

### 2.9. Measurement of IFN-γ Levels

Interferon-γ levels were measured in all serum samples collected on day 34 of the vaccination regimen using Rat IFN-**γ** Platinum the enzyme-linked immunosorbent assay; ELISA (eBioscience™ ELISA Kits—Thermo Fisher Scientific, Waltham, MA, USA) following manufacturer’s instructions. The concentration of INF-γ in each sample was determined by a standard curve and multiplied by the initial dilution factor. Levels of INF-γ were expressed as pg/mL.

### 2.10. PAGs Proteins Immunoblot Analysis

Immunochemical detection of PAGs on nitrocellulose paper was carried out as described previously [19]. Proteins were separated by sodium dodecylsulafte-polyacrylamide gel electrophoresis (SDS-PAGE) and electro-transferred to nitrocellulose membrane in a trans-blot cell (Bio-Rad). After blocking to minimize non-specific protein binding, the membrane was incubated with serum from PAGs-immunized rats. Bound antibody was detected by subsequent incubation with peroxidase-conjugated goat anti-rat IgG, followed by addition of enhanced chemiluminescent reagent (Amersham) and detection using X-ray film.

### 2.11. Isolation of Lipopolysaccharide (LPS) and Immunoblot Analysis

LPS was extracted from PA cells which had been cultured overnight on NB medium using the phenol–water method [20]. Cells were pretreated with proteinase K, DNase and RNase to get rid of proteins and nucleic acids, respectively. Final LPS preparation was dialyzed extensively against distilled water and then lyophilized.

SDS-PAGE has performed on 12% (*w*/*v*) polyacrylamide gels as previously described [21]. Following electrophoresis, LPS gels were subjected to Ammonia Silver staining using the Phast GelTH (Pharmacia, Stockholm, Sweden) kit. Other gels were stained with Coomassie blue R250 (Sigma, St. Louis, MI, USA).

From our experience, purified *P. aeruginosa* LPS does not transfer as efficiently from the SDS-polyacrylamide gels onto the nitrocellulose membrane as outer membrane proteins. LPS Samples were resolved on SDS-gradient polyacrylamide gels (EzWay PAG Pre-cast gel, 12%, Koma Biotech, Seoul, Korea) using an X-cell II™ Mini-Cell (Invitrogen, Carlsbad, CA, USA), and blotted onto nitrocellulose membranes in transfer buffer (25 mM Tris-Cl, pH 8.3, 192 mM glycine, 20% *v*/*v* methanol) for 1 h at 100 V. Membranes were blocked in 5% skim milk in phosphate-buffered saline (PBS, pH 7.3) containing 0.05% tween-20 (PBST), overnight, and probed with PAGs antibodies overnight. Bound antibody was detected by subsequent incubation with peroxidase-conjugated goat anti-rat IgG followed by chemiluminescence detection.

### 2.12. Statistical Analysis of Data

All data were expressed as mean ± standard deviation using GraphPad InStat version 3.10 for Windows, GraphPad Software, La Jolla, California, CA, USA. For normally distributed data, Student’s *t* test was used. Results were considered statistically significant when *p* value was less than 0.05.

## 3. Results

### 3.1. Evaluation of PAGs Quality

DNA and protein concentrations were quantified after each step during the PAGs preparation where PAGs prepared by Scheme 2 showed the least protein and DNA content after the final ethanol step (Table 2). Using agarose gel electrophoresis, DNA content was only identified in living bacteria and not in PAGs (Figure 2A). The entire destruction of genomic DNA in PAGs was revealed by agarose gel electrophoresis after various chemical treatments. Consequently, Scheme 2 was used to prepare PAGs for all subsequent experiments. SDS-PAGE showed that the protein content of PAGs is much lower than normal PA cells (Figure 2B) indicating the loss of cytoplasmic proteins in the case of PAGs.

After each step of PA ghost preparation, smears of cells were stained with crystal violet and compared with stained live PA under a light microscope (LM) (×100) (Figure 3A). External structure integrity was observed with PAGs indicating the success of the preparation procedure. The quality of the prepared PAGs was monitored, based on the bacterial surface structure, using Scanning Electron Microscope (SEM). Electron micrographs proved the correct three-dimensional (3D) structure of PAGs (Figure 3B). Moreover, SEM photos for PAGs revealed the pores resulting from the treatment with chemical agents where there were no gross changes in cellular morphology (Figure 3B).

### 3.2. Elicitation of the Specific Immune Response following Oral Immunization with PAGs

Serum agglutination titer was statistically significant in PAGs-immunized rats when compared with the negative control group (Figure 4A). Moreover, a significant increase in phagocytic and killing activity of neutrophils was observed in PAGs-immunized rats when compared with control rats (Figure 4B,C). Determination of serum IFN-γ concentration revealed elevated levels in immunized rats reflecting the stimulation of cell-mediated immunity (Figure 4D).

### 3.3. Bioburden in Different Organs of Vaccinated and Unvaccinated Rats after Challenging with P. aeruginosa

Vaccinated and unvaccinated rats were challenged orally, and organ bioburden was determined five days post-challenge. Immunized rats showed statistically significant lower bioburden in all tested organs (lungs, kidneys, liver, and gut) when compared with the unvaccinated group (Figure 5).

### 3.4. Oral Immunization with PAGs Protects against Infection of Artificial Ulcers in Diabetic Rats

Survival and ulcer healing of diabetic rats with infected ulcers was observed for 6 days where all animals of the control group succumbed to death by the fifth day. On the contrary, immunized rats showed complete vitality and survival which was statistically significant (Figure 6). In addition, PAGs-immunized rats showed almost complete healing of infected artificial ulcers on the sixth day of the challenge (Figure 7). Unvaccinated diabetic rats did not show significant ulcer healing when compared with the PAGs-immunized group (Figure 7).

### 3.5. PAGs-Specific Antibodies Interact with LPS and Proteins of P. aeruginosa

LPS was successfully extracted from PA cells and was checked by staining of SDS-PAGE gels with coomassie blue and silver stains. As expected, coomassie blue staining did not show any bands confirming the absence of any protein contaminant in the purified LPS (Figure 8A). On the contrary, bands of LPS were obvious upon silver staining (Figure 8B). Immunoblot analysis was subsequently performed where antibodies against PAGs obtained from orally immunized rats recognized fractions of the *P. aeruginosa* LPS pure extract (Figure 8C). Interaction between the rough core region of the electrophoretically separated LPS and the PAGs antibodies was clearly observed (Figure 8C).

When the prepared PAGs were analyzed by standard SDS-PAGE and Western blot, serum samples from orally immunized rats were reactive with protein bands of the prepared PAGs indicating the successful elicitation of PAGs-specific antibodies (Figure 8D,E).

## 4. Discussion

PA infections represent major clinical problems globally, particularly for critically ill and immunocompromised patients [2,22,23]. For almost half a century, many research attempts [6,23,24,25,26] have focused on development of a vaccine against infections caused by PA using capsular polysaccharide, LPS, iron acquisition proteins, outer membrane proteins (OMPs), or even heat-killed *P. aeruginosa* but no reliable vaccine is available so far.

This study is concerned with producing PAGs and investigating their ability to induce specific immune responses via the oral route of administration. The study followed the SLRP protocol for preparation of chemically induced BGs as previously described for *E. coli* by Amara and colleagues [11]. During preparation, bacterial cells were exposed to chemical compounds either alone or in certain combinations, and gentle centrifugation and washing steps were enabled to remove the DNA and the cytoplasmic protein content. The quality of the prepared PAGs was monitored using both light and electron microscope (EM). Scanning EM has proved the correct 3D structure of PAGs.

Treated BGss and non-treated control PA cells were analyzed for the remaining protein by SDS-PAGE analysis. As expected, non-treated cells showed strong protein bands while PAGs showed weaker protein bands due to exhaustion of cytoplasmic proteins. Moreover, degradation of bacterial genomic DNA has been observed in PAGs when compared with non-treated PA cells. Similar results were observed with ghosts of other bacteria prepared by the SLRP method which confirm the consistency and usefulness of this method [11,12,16].

Since PA is an opportunistic organism, we used a diabetic rat model to test for the vaccine efficacy. It is well-reported that PA is problematic in diabetic patients especially those with diabetic foot infections [2,3]. Alloxan-induced diabetic rats were immunized thrice via the oral route and the immune response was examined. Interestingly, oral immunization with PAGs resulted in systemic agglutinating antibodies and an increase in the phagocytic and killing activities of neutrophils in immunized rats. Moreover, the level of serum IFN-γ was significantly higher in PAGs-immunized rats than non-immunized ones. These results suggest the successful elicitation of both humoral and cell-mediated immune responses. Supporting our finding, an interferon-gamma (IFN-) release assay (IGRA) is used to assess a cell-mediated immune (CMI) response to *Mycobacterium bovis*, comparable to the tuberculin skin test [27]. Consequently, we used a rat diabetic ulcer model to evaluate PAGs vaccine efficacy. Intriguingly, PAGs orally immunized rats demonstrated reasonable healing rate and complete vitality and survival over the course of 6 days following ulcer PA infection. On the contrary, unvaccinated diabetic rats died by day 5 following ulcer infection most likely due to systemic spread of infection accompanied by lack of systemic immunity.

When the animals were challenged with a high oral dose of PA, the bioburden of kidneys, lungs, livers, and gut was significantly lower in PAGs-immunized rats. This further confirms the protective vaccine efficacy of PAGs where it comparatively limited the systemic spread of PA. Importantly, PA is reported to translocate from gut to cause systemic infections in immunocompromised patients [28,29,30]. This fact let us use the oral route in bacterial challenge to investigate the protective vaccine efficacy of oral PAGs to prevent such systemic spread. Indeed, Toth and colleagues found that oral vaccination with a live attenuated *Salmonella dublin* strain expressing the outer membrane protein I of PA prevented the translocation of PA from gut upon oral challenge [30]. Our work using BGs has obviously a superior safety profile compared to live attenuated vaccines. It is noteworthy that in our oral challenge model the overall organ bioburden was high in both unvaccinated and vaccinated rats. Concomitant administration of oral immunoadjuvants [31,32,33] is a potential future approach which could improve PAGs protective vaccine efficacy upon oral administration.

Cripps and co-workers previously demonstrated that paraformaldehyde-killed PA given by the oral route as well as subcutaneous route decreased lung bioburden following intra-tracheal PA challenge [26]. Our results concerning PAGs prepared by the SLRP corroborate that the oral immunization route is useful. It is noteworthy that we immunized the rats orally only three times with PAGs while Cripps and colleagues used 10 oral doses of killed PA to immunize the rats [26]. The preserved 3D structure of PAGs is believed to result in better immunogenicity and vaccine efficacy. Furthermore, the use of the oral immunization route has the advantages of convenience and safety over parenteral administration, especially for children. That is why several investigators used the oral route in vaccination studies [34,35,36,37].

In the present study, immunoblotting analysis was performed to determine the antigen specificity of PAGs-induced antibodies. The results showed that serum antibodies from orally PAGs-immunized rats recognized both LPS and protein fractions of PA. It is known that the LPS of Gram-negative bacteria is pyrogenic upon injection in humans [38]. However, since we used the oral administration route, this pyrogenic effect is irrelevant and the presence of LPS in PAGs further augments the immunogenicity of the orally administered vaccine. Nevertheless, it was previously reported that the LPS presence in parenterally administered BGs showed minimal toxicity and thus does not limit their parenteral vaccine use [39].

The use of non-living ghost bacteria in our study has the advantage of ensuring safety over previously reported live attenuated vaccine vectors [36,40]. We recently reported that *A. baumannii* ghost (ABG), prepared by the SLRP method, was able to induce humoral and cellular immunity and provide protection against live bacteria [16]. ABG was administered to rats via different administration routes. It was noticed that ABG-vaccinated rats, regardless of the administration route, showed marked protection against virulent *A. baumannii*. On the other hand, a 100% mortality rate was observed in non-vaccinated rats.

Oral delivery of BGs (EHEC strain O157:H7) was found to be a successful approach for future vaccine use because it mimics natural bacterial infection [41]. Non-living whole-cells preparations such as BGs or subunit bacterial vaccines, on the other hand, have been proven to be less immunogenic when given orally [42]. Meanwhile, Mayr et al. (2005) claimed that stomach acid and proteolytic enzymes can degrade the quality of BGs given orally, affecting the level of immune response required [37]. To generate protective immunity, a mucosal adjuvant can be added to these vaccination formulations [43]. Cimetidine (50 mg/kg) administered intravenously to lower gastric acid output in the stomach (in rabbit) and 15 mL of 5% sodium bicarbonate injected twice intragastrically to neutralize stomach acid were utilized by Xian et al. (2020) [44]. For example, cellulose acetate phthalate (Eudragit) is frequently employed as a polymer film to protect a capsulated vaccination since it is insoluble at low pH in the stomach but dissolves rapidly at higher pH in the gut, depending on the type of Eudragit used [45,46]. However, increasing the number of BGs utilized and using successful repeated boosters that allow for proper immune response are still the favored approaches [12,16]. In the present study, the created PAGs retained the external structural features of the Gram-negative bacterium PA. This implies the possible cross-protection against other Gram-negative bacteria due to conserved basic structural cell wall components. Indeed, the PAGs cross-protection against other bacteria is worthy of further investigation.

Overall, this study is considered the first to develop chemically induced bacterial ghost for *P. aeruginosa* using the critical chemical concentration of NaOH, SDS, CaCO_3_, and H_2_O_2_ where the MIC and MGC of these substances were capable of producing efficient PAGs able to induce specific immune response against PA. In addition, this study highlights the effectiveness of the oral immunization route as a successful way to vaccinate against *P. aeruginosa* infection in diabetic rats using PAGs. The ghost vaccine not only induced powerful humoral and cell-mediated immune responses among vaccinated rats, but it also provided significant protection systemically and in diabetic ulcer infection model. These findings demonstrate the potential of *P. aeruginosa* ghosts as an effective vaccine and warrant their further testing in other animal models such as respiratory, sepsis, burn, and urinary tract infection models.

## Figures and Tables

**Figure 1 vaccines-10-00910-f001:**
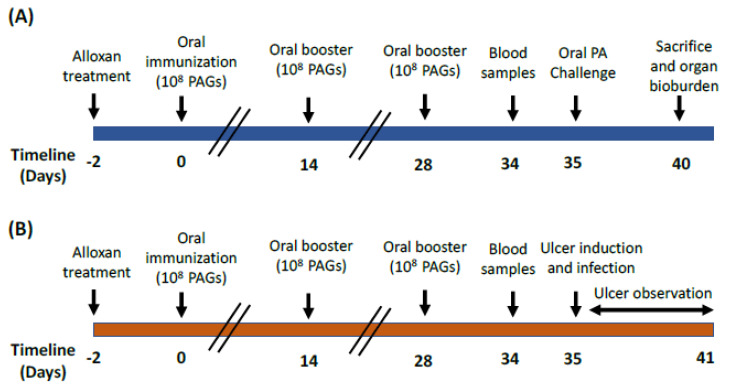
An illustration of the experimental design on animals. Experimental protocols are shown for rats challenged orally (**A**) or through ulcer induction and infection (**B**). Negative control groups (not shown) were also included and were given saline instead of the vaccine. Each experimental group included six rats. Blood samples were collected on day 34 to determine the agglutinating antibody titers, phagocytic and killing activities, and serum IFN-γ levels. The timeline is not to scale.

**Figure 2 vaccines-10-00910-f002:**
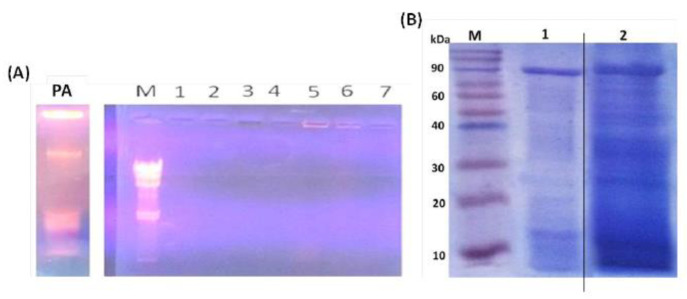
Characterization of Pseudomonas aeruginosa ghosts (PAGs). (**A**) Agarose gel electrophoresis of prepared PAGs. Lane PA: *P. aeruginosa* cells showing intense DNA bands; Lane M: DNA marker; Lanes 1–4: *P*. *aeruginosa* ghost prepared by Scheme 1; Lanes 5–7: *P. aeruginosa* ghost prepared by Scheme 2. (**B**) SDS-PAGE of prepared PAGs. M: Protein ladder; Lane 1: PAGs preparation; Lane 2: *P. aeruginosa* cells. Samples were boiled in sample buffer for 3 min and loaded into gel.

**Figure 3 vaccines-10-00910-f003:**
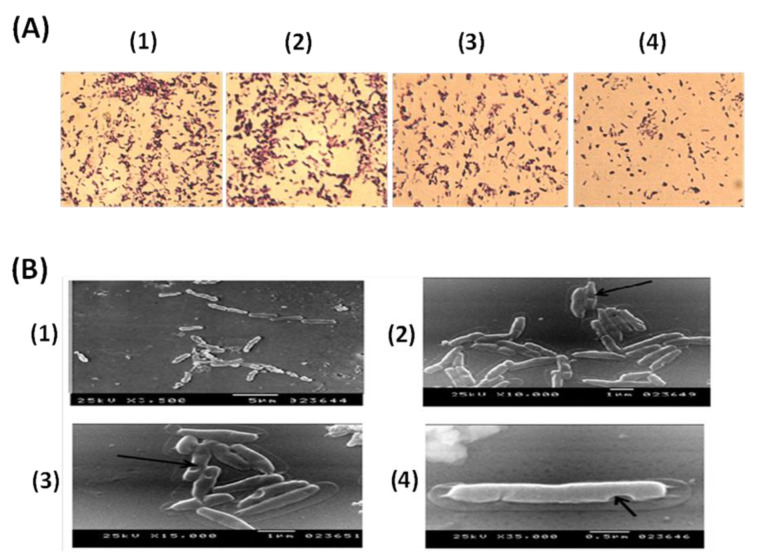
Microscopical examination of PAGs. (**A**) Light microscope pictures following crystal violet simple staining using oil immersion lens (×100): (**1**) *P. aeruginosa* untreated cells, (**2**) cells after the basic step, (**3**) cells after the H_2_O_2_ step, and (**4**) *P. aeruginosa* ghost after the ethanol step. External structure integrity is observed in all smears. (**B**) Scanning electron micrographs at different magnification powers. (**1**) Normal *P. aeruginosa* cells showing intact cell walls. (**2**), (**3**), and (**4**) show BGs after chemical treatment showing a 3D non-deformed cell wall where the arrows show the pores formed.

**Figure 4 vaccines-10-00910-f004:**
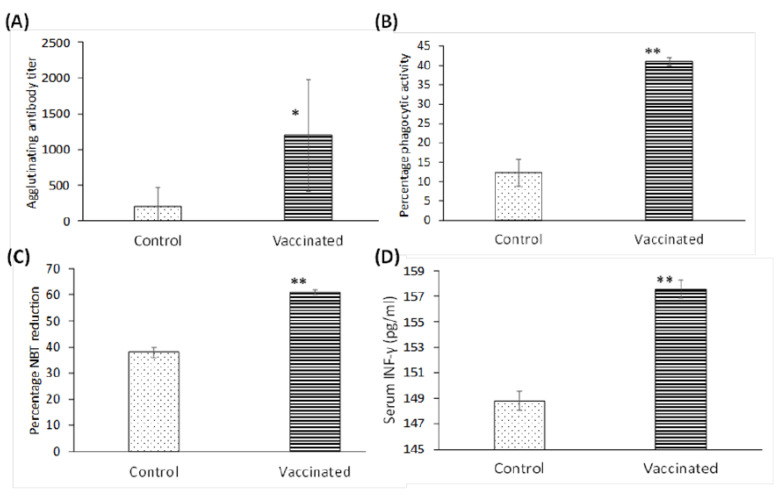
PAGs increase immunological responses in diabetic rats after oral vaccination. (**A**) Agglutinating antibody titers as determined by the slide agglutination test. Percentage phagocytic (**B**) and NBT reduction (**C**) activity was also evaluated. (**D**) Serum IFN-γ was determined by ELISA. All tests were carried out on samples recovered on day 34 of the vaccination regimen. All experiments were repeated twice. (*) *p* < 0.05; (**) *p* < 0.001 using Mann–Whitney test (**A**) and Student’s *t* test (**B**–**D**) (N = 6).

**Figure 5 vaccines-10-00910-f005:**
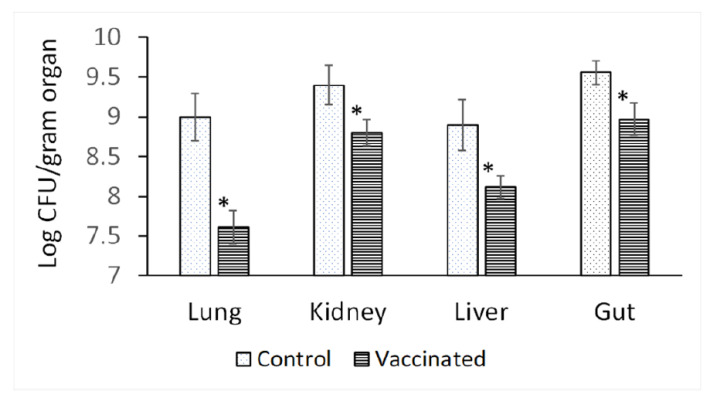
The bioburden of organs in challenged rats. Diabetic animals were orally vaccinated with 10^8^ PAGs for a total of three doses at two-week intervals. On day 35, both vaccinated and unvaccinated groups were challenged orally with 10^8^ CFU of *P. aeruginosa*, and organ bioburden was determined 5 days post-challenge. (*) *p* < 0.05 using Student’s *t* test (N = 6).

**Figure 6 vaccines-10-00910-f006:**
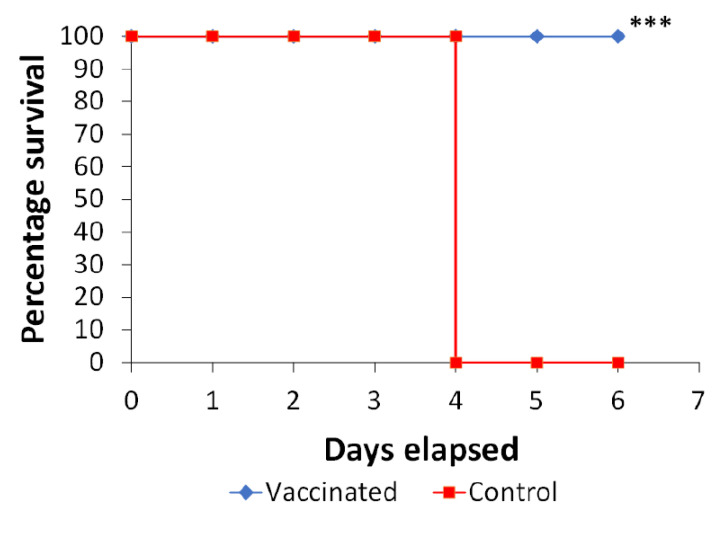
Following PA infection of artificial ulcers, a Kaplan–Meier survival curve was created. Artificial ulcers of PAGs orally immunized diabetic rats were infected with PA and animal survival was observed for 6 days. (***) *p* < 0.001 by Log-rank (Mantel–Cox) and Gehan–Breslow–Wilcoxon tests.

**Figure 7 vaccines-10-00910-f007:**
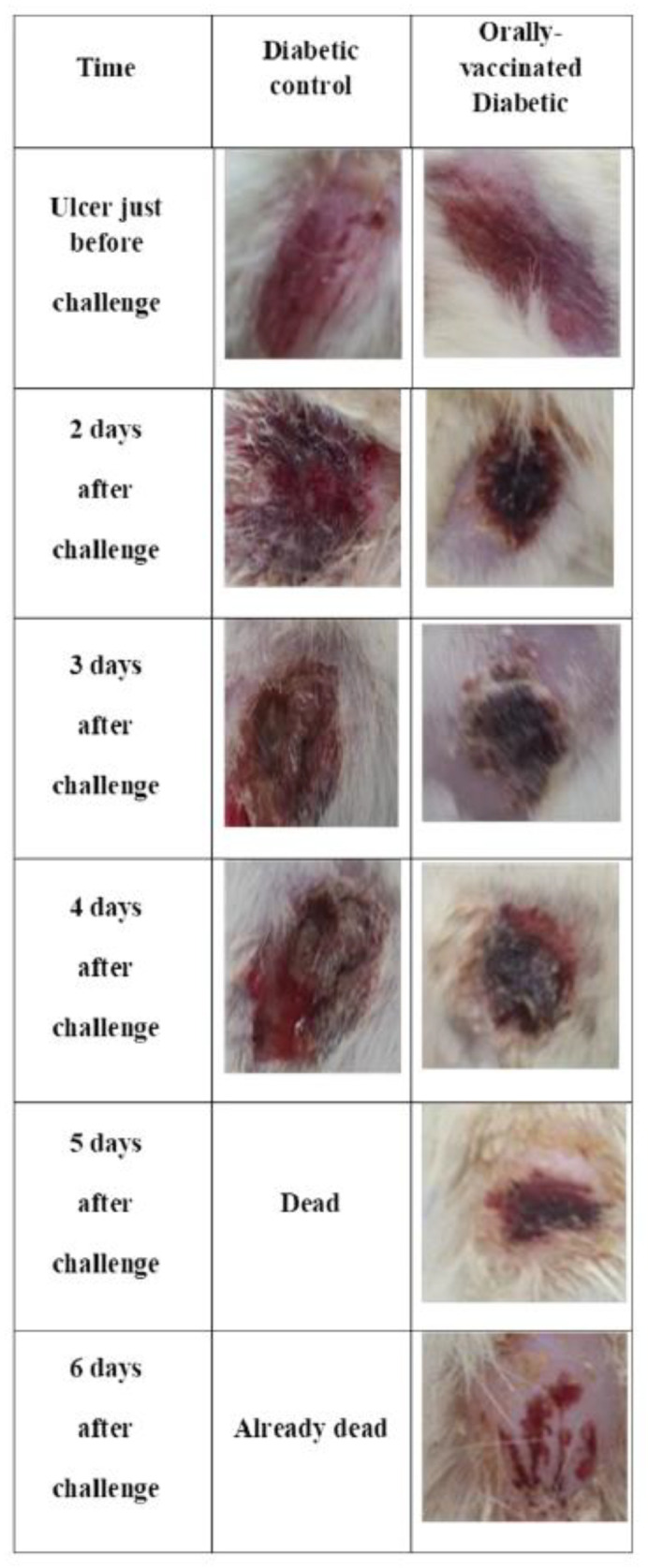
The rate of ulcer healing in various PA-infected animal groups. Artificial ulcers in vaccinated and unvaccinated diabetic rats were infected with PA and healing was observed for 6 days. All unvaccinated rats died by the fifth day post-challenge.

**Figure 8 vaccines-10-00910-f008:**
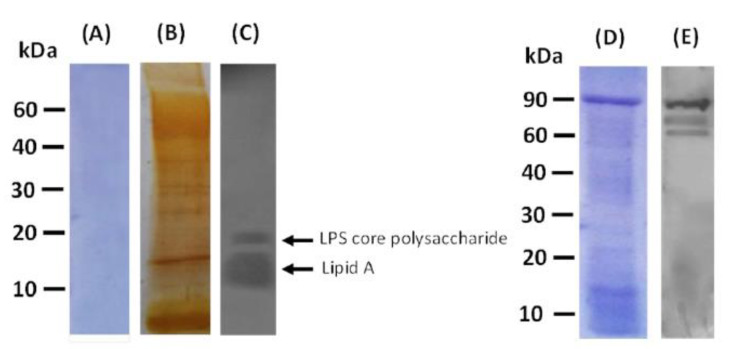
Immune responsiveness of PAGs antibodies to LPS and proteins from P. aeruginosa. (**A**) Coomassie blue-stained gel of the purified extracted *P. aeruginosa* LPS. (**B**) The silver-stained gel of purified LPS. (**C**) Western blot of purified LPS using serum from PAGs immunized rats as the primary antibody. (**D**) Coomassie blue-stained gel of prepared PAGs. (**E**) Western blot against PAGs proteins using serum from PAGs-immunized rats as the primary antibody.

**Table 1 vaccines-10-00910-t001:** Experimental schemes used for preparation of PAGs.

	Used Chemical Concentration			
Experimental Scheme	NaOH	CaCO_3_	SDS	H_2_O_2_
Scheme 1	MGC	1.05 µg /mL	MIC	MIC
Scheme 2	MGC	0.35 µg /mL	MIC	MGC

**Table 2 vaccines-10-00910-t002:** Protein and DNA content of prepared PAGs.

	Protein and DNA Content (µg/mL) after Each Step					
	Basic Step		H_2_O_2_ Step		Ethanol Step	
	Protein	DNA	Protein	DNA	Protein	DNA
PAGs prepared by Scheme 1	1720.08	130.9	280.16	27.14	102.35	25.34
PAGs prepared by Scheme 2	1430.35	163.17	154.98	43.54	67.08	13.02

## Data Availability

Publicly available datasets were analyzed and included in this study.

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
