# Peer review of "Bacterial Ghosts of *Pseudomonas aeruginosa* as a Promising Candidate Vaccine and Its Application in Diabetic Rats"

_vaccines, 2022, doi:10.3390/vaccines10060910_

Round 1
Reviewer 1 Report
Minor comments
- Line 16: ‘aeruginosa’ change to italic style
- Line 19: “Pseudomonas aeruginosa’ change to italic style
- Line 89: after MIV, delete ‘)’
- Line 183: after ‘as an indicator for T helper type 1 (Th1) immune response” cite the reference
- Line 209: ‘Samples’ should be samples.
- Line 323: change ‘PA ghosts’ to PAGs to keep consistency throughout the text
- Line 325: change ‘bacterial ghosts’ to BGs to keep consistency throughout the text
Major comments
- Figure 2B : It is hard to compare protein profiles between PAGs and wild-type PA bacteria due to low resolution. Authors need a better SDS-PAGE picture.
- Figures 8D and 8E: The authors showed that PAGs-specific antibodies react with PAGs proteins. They missed a negative control in the analysis. Therefore, they need to test Western blot analysis between PAGs-specific antibodies and wild-type PA bacteria proteins.
- Lines 346-347: The authors described that their results supported successful elicitation of the cell-mediated immune response. Of course, IFN-γ is a main contributor to the cell-mediated immune response via macrophage activation, the increasing of antigen presentation via the MHC class I and II pathways immune responses. To prove cell-mediated response, authors are better to provide helper T cell population by FACS analysis. If they have difficulties doing FACS, authors need to describe some sentences about IFN-γ involved in cell-mediated immune response from previous reports with references.
Author Response
Responses to Reviewer No.1
Minor comments
- Line 16: ‘aeruginosa’ change to italic style: Corrected
- Line 19: “Pseudomonas aeruginosa’ change to italic style corrected Corrected
- Line 89: after MIV, delete ‘)’ Corrected
- Line 183: after ‘as an indicator for T helper type 1 (Th1) immune response” cite the reference. . “as an indicator for T helper type 1 (Th1) immune response” has been deleted and corrected in the text.
- Line 209: ‘Samples’ should be samples
- Line 323: change ‘PA ghosts’ to PAGs to keep consistency throughout the text Corrected
- Line 325: change ‘bacterial ghosts’ to BGs to keep consistency throughout the text Corrected
Major comments
- Figure 2B : It is hard to compare protein profiles between PAGs and wild-type PA bacteria due to low resolution. Authors need a better SDS-PAGE picture.
Authors’ Response: The protein bands of Bacteria are clear, but its ghost usually appeared like this because of different treatments with chemical agent during PAGs preparation. Unfortunately this is the best resolution we have got.
- Figures 8D and 8E: The authors showed that PAGs-specific antibodies react with PAGs proteins. They missed a negative control in the analysis. Therefore, they need to test Western blot analysis between PAGs-specific antibodies and wild-type PA bacteria proteins.
Authors’ Response: In our previous published papers, we have not included such negative control. Therefore, this study was designed as our previous work. Your point is very considerable and will be considered in future works. Please see.
- Sheweita SA, Batah AM, Ghazy AA, Hussein A, Amara AA. A new strain of Acinetobacter baumannii and characterization of its ghost as a candidate vaccine. J Infect Public Health. 2019 Nov-Dec;12(6):831-842.;
- Hussein KE, Bahey-El-Din M, Sheweita SA. Immunization with the outer membrane proteins OmpK17 and OmpK36 elicits protection against Klebsiella pneumoniae in the murine infection model. Microb Pathog. 2018 Jun;119:12-18.
- Amro AA, Neama AJ, Hussein A, Hashish EA, Sheweita SA. Evaluation the surface antigen of the Salmonella typhimurium ATCC 14028 ghosts prepared by "SLRP". ScientificWorldJournal. 2014 Mar 19;2014:840863. doi: 10.1155/2014/840863
- Lines 346-347: The authors described that their results supported successful elicitation of the cell-mediated immune response. Of course, IFN-γ is a main contributor to the cell-mediated immune response via macrophage activation, the increasing of antigen presentation via the MHC class I and II pathways immune responses. To prove cell-mediated response, authors are better to provide helper T cell population by FACS analysis. If they have difficulties doing FACS, authors need to describe some sentences about IFN-γ involved in cell-mediated immune response from previous reports with references. Authors’ Response: We do not have FACS analysis to include more data. A new sentence about IFN-γ involved in cell-mediated immune response has been included in the text [Supporting our finding, an interferon-gamma (IFN-) release assay (IGRA) is used to assess a cell-mediated immune (CMI) response to Mycobacterium bovis, comparable to the tuberculin skin test [Smith et al., 2021 reference number 27]..Please see page 349-351.
Responses to Reviewer No.1
Minor comments
- Line 16: ‘aeruginosa’ change to italic style: Corrected
- Line 19: “Pseudomonas aeruginosa’ change to italic style corrected Corrected
- Line 89: after MIV, delete ‘)’ Corrected
- Line 183: after ‘as an indicator for T helper type 1 (Th1) immune response” cite the reference. . “as an indicator for T helper type 1 (Th1) immune response” has been deleted and corrected in the text.
- Line 209: ‘Samples’ should be samples
- Line 323: change ‘PA ghosts’ to PAGs to keep consistency throughout the text Corrected
- Line 325: change ‘bacterial ghosts’ to BGs to keep consistency throughout the text Corrected
Major comments
- Figure 2B : It is hard to compare protein profiles between PAGs and wild-type PA bacteria due to low resolution. Authors need a better SDS-PAGE picture.
Authors’ Response: The protein bands of Bacteria are clear, but its ghost usually appeared like this because of different treatments with chemical agent during PAGs preparation. Unfortunately this is the best resolution we have got.
- Figures 8D and 8E: The authors showed that PAGs-specific antibodies react with PAGs proteins. They missed a negative control in the analysis. Therefore, they need to test Western blot analysis between PAGs-specific antibodies and wild-type PA bacteria proteins.
Authors’ Response: In our previous published papers, we have not included such negative control. Therefore, this study was designed as our previous work. Your point is very considerable and will be considered in future works. Please see.
- Sheweita SA, Batah AM, Ghazy AA, Hussein A, Amara AA. A new strain of Acinetobacter baumannii and characterization of its ghost as a candidate vaccine. J Infect Public Health. 2019 Nov-Dec;12(6):831-842.;
- Hussein KE, Bahey-El-Din M, Sheweita SA. Immunization with the outer membrane proteins OmpK17 and OmpK36 elicits protection against Klebsiella pneumoniae in the murine infection model. Microb Pathog. 2018 Jun;119:12-18.
- Amro AA, Neama AJ, Hussein A, Hashish EA, Sheweita SA. Evaluation the surface antigen of the Salmonella typhimurium ATCC 14028 ghosts prepared by "SLRP". ScientificWorldJournal. 2014 Mar 19;2014:840863. doi: 10.1155/2014/840863
- Lines 346-347: The authors described that their results supported successful elicitation of the cell-mediated immune response. Of course, IFN-γ is a main contributor to the cell-mediated immune response via macrophage activation, the increasing of antigen presentation via the MHC class I and II pathways immune responses. To prove cell-mediated response, authors are better to provide helper T cell population by FACS analysis. If they have difficulties doing FACS, authors need to describe some sentences about IFN-γ involved in cell-mediated immune response from previous reports with references. Authors’ Response: We do not have FACS analysis to include more data. A new sentence about IFN-γ involved in cell-mediated immune response has been included in the text [Supporting our finding, an interferon-gamma (IFN-) release assay (IGRA) is used to assess a cell-mediated immune (CMI) response to Mycobacterium bovis, comparable to the tuberculin skin test [Smith et al., 2021 reference number 27]..Please see page 349-351.
Responses to Reviewer No.1
Minor comments
- Line 16: ‘aeruginosa’ change to italic style: Corrected
- Line 19: “Pseudomonas aeruginosa’ change to italic style corrected Corrected
- Line 89: after MIV, delete ‘)’ Corrected
- Line 183: after ‘as an indicator for T helper type 1 (Th1) immune response” cite the reference. . “as an indicator for T helper type 1 (Th1) immune response” has been deleted and corrected in the text.
- Line 209: ‘Samples’ should be samples
- Line 323: change ‘PA ghosts’ to PAGs to keep consistency throughout the text Corrected
- Line 325: change ‘bacterial ghosts’ to BGs to keep consistency throughout the text Corrected
Major comments
- Figure 2B : It is hard to compare protein profiles between PAGs and wild-type PA bacteria due to low resolution. Authors need a better SDS-PAGE picture.
Authors’ Response: The protein bands of Bacteria are clear, but its ghost usually appeared like this because of different treatments with chemical agent during PAGs preparation. Unfortunately this is the best resolution we have got.
- Figures 8D and 8E: The authors showed that PAGs-specific antibodies react with PAGs proteins. They missed a negative control in the analysis. Therefore, they need to test Western blot analysis between PAGs-specific antibodies and wild-type PA bacteria proteins.
Authors’ Response: In our previous published papers, we have not included such negative control. Therefore, this study was designed as our previous work. Your point is very considerable and will be considered in future works. Please see.
- Sheweita SA, Batah AM, Ghazy AA, Hussein A, Amara AA. A new strain of Acinetobacter baumannii and characterization of its ghost as a candidate vaccine. J Infect Public Health. 2019 Nov-Dec;12(6):831-842.;
- Hussein KE, Bahey-El-Din M, Sheweita SA. Immunization with the outer membrane proteins OmpK17 and OmpK36 elicits protection against Klebsiella pneumoniae in the murine infection model. Microb Pathog. 2018 Jun;119:12-18.
- Amro AA, Neama AJ, Hussein A, Hashish EA, Sheweita SA. Evaluation the surface antigen of the Salmonella typhimurium ATCC 14028 ghosts prepared by "SLRP". ScientificWorldJournal. 2014 Mar 19;2014:840863. doi: 10.1155/2014/840863
- Lines 346-347: The authors described that their results supported successful elicitation of the cell-mediated immune response. Of course, IFN-γ is a main contributor to the cell-mediated immune response via macrophage activation, the increasing of antigen presentation via the MHC class I and II pathways immune responses. To prove cell-mediated response, authors are better to provide helper T cell population by FACS analysis. If they have difficulties doing FACS, authors need to describe some sentences about IFN-γ involved in cell-mediated immune response from previous reports with references. Authors’ Response: We do not have FACS analysis to include more data. A new sentence about IFN-γ involved in cell-mediated immune response has been included in the text [Supporting our finding, an interferon-gamma (IFN-) release assay (IGRA) is used to assess a cell-mediated immune (CMI) response to Mycobacterium bovis, comparable to the tuberculin skin test [Smith et al., 2021 reference number 27]..Please see page 349-351.

Reviewer 2 Report
The study is interesting enough and it covers very important medical problem. The content is presented clearly. Different experiments have been performed trying to evaluate immune response after the vaccination of animals. I have few concerns which want to be addressed:
- In Figure 3, part B we see the pictures of PA, however the picture B1 is presented in lower magnification than B2, B3 and B4. It is impossible to see the surface in normal PA cells and to compare it with bacterial hosts cells;
- In Figure 3, part A we see the pictures where the amount of bacteria looks less in A4 than in other pictures. It should be explained why the numbers of bacteria became lower going from the untreated sample to the treated samples. Are some of the cells were lysed during the treatment process?
- More exact technique how do the rats were vaccinated (how the suspension was given) should be explained in the section 2.3 (rows 111-122).
- Pictures in Figure 2 are of poor quality. For example, it is almost impossible to see differences between B1 and B2. Maybe it is possible a bit improve pictures quality (to add better contrast)?
- One of the main concern is about the time of the experiment. For me it looks too short. Why survived animals were observed only for two days after death of the control animals? We see that PA were presented in quite high numbers in both groups of animals. In the experimental group the amount of PA in organs were lower but overall this bacterium were presented and probably it can be inhibited for 6-7 days after challenge therefore we may expect that the clinical symptoms or even death of animals may become later. However, we do not see such data. If the vaccinated rats survived I would recommend to mention this in the figure or at least in the text (for example rats were observed at least for 2 weeks after the challenge).
- In Figure 7 we see appropriate pictures of wounds however, the single animals from the groups are presented. Where are the pictures of other animals/ulcers?
- Figure 5 presents organ bioburden and we see high numbers of PA in different organs in vaccinated animals. Knowing the fast reproduction of bacteria we may expect that the numbers may change vary fast. It looks that organs are not protected from PA infection and the immunity does not guarantee sterile immunity; this probably should be discussed. It would be useful to see bioburden of vaccinated animals in more prolonged time i.e to see the dynamics of bioburden.
- In the section Discussion it would be useful to see short discussion about the possible vaccine interaction with stomach acid etc. i.e. to discuss how do the vaccine remains non affected by physiological processes in the body. What are the data from other studies?
- I would suggest to change the title of the manuscript more focusing on the experiment made rather to make title from two parts (the initial part looks like a review article rather than original article based on experiments). But it is accept to the authors.
- In some parts of the results (rows 226-227; 242-247; 275-276) there are methods described, not the results therefore this text should be moved to the methods in case it is still not presented in the section Methods.
- In the heading 3.3 the word "protection" for me looks not correct. I suggest to change this heading to be more objective as the vaccination resulted in lower bioburden but not guaranteed protection against PA.
- The sentence in row 88 looks unfinished. The text should be reorganized the way that the sentence means finished statement.
- Please italize P. aeruginosa in the abstract.
Author Response
Responses to Reviewer No.2
The study is interesting enough and it covers very important medical problem. The content is presented clearly. Different experiments have been performed trying to evaluate immune response after the vaccination of animals. I have few concerns which want to be addressed:
- In Figure 3, part B we see the pictures of PA, however the picture B1 is presented in lower magnification than B2, B3 and B4. It is impossible to see the surface in normal PA cells and to compare it with bacterial hosts cells. Authors’ Response: The effect of various treatments on the PA may be seen in images B1 through B4. The cell membrane of B4 was clearly opened because of the treatments.
- In Figure 3, part A we see the pictures where the amount of bacteria looks less in A4 than in other pictures. It should be explained why the numbers of bacteria became lower going from the untreated sample to the treated samples. Are some of the cells were lysed during the treatment process?
Authors’ Response: This difference is merely due to sample dilution during the repeated steps of ghost preparation.
- More exact technique how do the rats were vaccinated (how the suspension was given) should be explained in the section 2.3 (rows 111-122).
Authors’ Response: “The rats were vaccinated orally using blunt-ended oral gavage needle.”. This statement has been added to the manuscript. Please see line 129
- Pictures in Figure 2 are of poor quality. For example, it is almost impossible to see differences between B1 and B2. Maybe it is possible a bit improve pictures quality (to add better contrast)?. Authors’ Response: At this point, it is quite difficult to undertake further experimental work because our student completed her Master's degree in 2017 and I am currently residing abroad of my home country.
- One of the main concern is about the time of the experiment. For me it looks too short. Why survived animals were observed only for two days after death of the control animals? We see that PA were presented in quite high numbers in both groups of animals. In the experimental group the amount of PA in organs were lower but overall this bacterium were presented and probably it can be inhibited for 6-7 days after challenge therefore we may expect that the clinical symptoms or even death of animals may become later. However, we do not see such data. If the vaccinated rats survived I would recommend to mention this in the figure or at least in the text (for example rats were observed at least for 2 weeks after the challenge).
Authors’ Response: We thank the reviewer for this note. In fact, the objective of this experiment was to examine protection against infection of ulcer and examination of ulcer healing rate rather than the survival of the animals themselves. We were surprised of the rapid death of unvaccinated animals but we measured the outcome of ulcer healing rather than death of vaccinated rats. The rats were healthy and had healed ulcers after the stated period.
- In Figure 7 we see appropriate pictures of wounds however, the single animals from the groups are presented. Where are the pictures of other animals/ulcers? Authors’ Response: Because it was impossible to include all the images, we chose one to represent each group.
- Figure 5 presents organ bioburden and we see high numbers of PA in different organs in vaccinated animals. Knowing the fast reproduction of bacteria we may expect that the numbers may change vary fast. It looks that organs are not protected from PA infection and the immunity does not guarantee sterile immunity; this probably should be discussed. It would be useful to see bioburden of vaccinated animals in more prolonged time i.e to see the dynamics of bioburden.
Authors’ Response: We thank the reviewer for this important comment. Indeed, there was a high count in organs of vaccinated rats, though significantly lower than non-vaccinated rats. The objective of the study is to improve the concept of using PAGs as a vaccine candidate. Achieving sterile immunity is not easy and this might be fulfilled using strong immunoadjuvants along with PAGs and using enteric coated tablets to ensure high dose release of PAGs in the intestine to maximize immune stimulation.
- In the section Discussion it would be useful to see short discussion about the possible vaccine interaction with stomach acid etc. i.e. to discuss how do the vaccine remains non affected by physiological processes in the body. What are the data from other studies?. Authors’ Response: Please see lines 396-410, a new paragraph is added to the discussion as you suggested.
- I would suggest to change the title of the manuscript more focusing on the experiment made rather to make title from two parts (the initial part looks like a review article rather than original article based on experiments). But it is accept to the authors. Authors’ Response: I think the title reflected the manuscript's content.
- In some parts of the results (rows 226-227; 242-247; 275-276) there are methods described, not the results therefore this text should be moved to the methods in case it is still not presented in the section Methods. Authors’ Response: Please see lines 95-98, 105-109; 126-132.
- In the heading 3.3 the word "protection" for me looks not correct. I suggest to change this heading to be more objective as the vaccination resulted in lower bioburden but not guaranteed protection against PA. Authors’ Response: It has been changed into “Bioburden in different organs of vaccinated and unvaccinated rats after challenging with P. aeruginosa”” Please see lines 274-275
-The sentence in row 88 looks unfinished. The text should be reorganized the way that the sentence means a finished statement. Authors’ Response: It has been corrected. Please see lines 8-90
- Please italize P. aeruginosa in the abstract. Authors’ Response: Corrected in the abstract.

Round 2
Reviewer 2 Report
The manuscript was improved however, as I understand at the moment not all improvements are possible.
The question about more images that are now in Figure 7 remained unanswered, because more pictures are possible to add as supplementary files and not necessary to put them into the text of the manuscript or into the Figure 7.
Some text describing methods (point 10) was not removed from the section Results. In this section results should be described but not the methods.
Author Response
Responses to the second round of revision
The question about more images that are now in Figure 7 remained unanswered, because more pictures are possible to add as supplementary files and not necessary to put them into the text of the manuscript or into the Figure 7. Please see supplementary file.
Some text describing methods (point 10) was not removed from the section Results. In this section results should be described but not the methods. Please see RED text in both Methods and Results sections.
